# Experiencing the Unprecedented COVID-19 Lockdown: Abu Dhabi Older Adults’ Challenges and Concerns

**DOI:** 10.3390/ijerph182413427

**Published:** 2021-12-20

**Authors:** Masood A. Badri, Mugheer A. Alkhaili, Hamad Aldhaheri, Guang Yang, Muna Albahar, Asma Alrashdi, Bushra Almulla, Layla Alhyas

**Affiliations:** 1Department of Community Development, Ministries Complex, Abu Dhabi CFJ7+2P, United Arab Emirates; Mugheer@addcd.gov.ae (M.A.A.); Hamad.aldhaheri@addcd.gov.ae (H.A.); Guang.yang@addcd.gov.ae (G.Y.); muna.albahar@addcd.gov.ae (M.A.); Asma.alrashdi@addcd.gov.ae (A.A.); Bushra.almulla@addcd.gov.ae (B.A.); Layla.Alhyas@addcd.gov.ae (L.A.); 2Department of Business Administration, United Arab Emirates University, Abu Dhabi 6M2H+44, United Arab Emirates

**Keywords:** older adults, COVID-19, social isolation, technological challenges, Abu Dhabi

## Abstract

This study focused on older adults (60+ years old) of both genders in Abu Dhabi during the COVID-19 pandemic before vaccines were made available (age ranged from 60 years to 75 years). They faced more strict rules of movement restriction and isolation that might have resulted in certain psychological feelings and social reactions. The main objective was to understand Abu Dhabi older adults’ psychological feelings during the pandemic and to identify their main concerns and challenges considering the various COVID-19-related policies and restrictions. The psychological feelings focused on fear, loneliness, sadness, irritability, emotional exhaustion, depressive symptoms, sleeping disorders, overeating, and excessive screen use. The objectives also included the changes in the psychological feelings concerning time. Other objectives covered better understanding the differences in (some activities) compared to the other age categories. Data were gathered through an online survey of community members from February to July 2020 as part of government initiatives (Department of Community Development). Responses were collected from 574 older adults in Abu Dhabi (60.1% male and 39.9% female). The analysis mainly used descriptive analysis, *t*-tests, analysis of variance (ANOVA), and simple trend analysis. For all tests, a *p*-value less than 0.05 was used for significance. The results pointed to the significant rise in feelings related to excessive screen use, fear, loneliness, and stress. The most significant concerns were related to more restrictions being imposed and not being able to see the grandchildren.The impact of new technologies on their quality of life was significantly reflected by respondents. The influence of the pandemic on older adults’ health and weight was also investigated. Analysis of variance, *t*-tests, and regression analysis with relevant tests were employed. The relevant results showed that some negative psychological feelings were common among older adults during the pandemic. However, the psychological feelings did not portray significant changes with time, except for sleeping disorders and overeating. Overall, older adults scored significantly different from other age groups on many challenges, concerns, and views regarding new technologies during the pandemic. No significant differences were observed regarding gender and marital status for the challenges and concerns. The research summarizes some policy guidance while noting some limitations of this study and future research directions.

## 1. Introduction

The World Health Organization (WHO) has identified older adults as especially vulnerable to the novel coronavirus outbreak [1]. Moreover, the COVID-19 virus has created complications, showing more fatal implications among older adults [2]. Meanwhile, the United Nations in May 2020 published a report warning that the COVID-19 pandemic is causing untold fear and suffering for older people across the world [3]. As the virus spreads rapidly and health and social protection systems get overwhelmed, older people may increasingly face vulnerability, abuse, and neglect, which should become the focus of policy considerations. Moreover, mortality data from the Oxford COVID-19 Evidence Service [4] indicated a risk of mortality of 3.6% for people in their 60s. The data also reported that the percentage increases to 8.0% and 14.8% for people in their 70s and 80s, respectively. 

Most countries issued regulations that encouraged people to stay at home and avoid contact with other people, possibly for an extended time, to shield older adults. There were also enforced lockdowns and curfews. Therefore, the global recommendation for the older population has stressed the issue of social isolation [5,6]. In its more general context, social isolation may include avoiding social contact with family members and friends, social distancing, and organizing the delivery of essential goods and items such as groceries and medications.

The first case of COVID-19 in the UAE was recorded on 29 January 2020 [7]. The UAE was one of the early countries to record confirmed cases. As a result, the UAE government organized an aggressive vaccination program. Such a dynamic vaccination program has effectively controlled the pandemic, as the number of COVID-19 infections fell by 62% from January 2021 to August 2021. New infections reported in the UAE followed a downward trend for more than two weeks since early August. To keep the pandemic at bay, the government extended the vaccination to all, as it began to provide a booster dose for specific groups as needed [8]. 

The government issued policies and rules concerning older adults during the COVID-19 pandemic as it progressed [8,9]. The rules and policies covered many aspects of everyday living, including, but not limited to, isolation, limits of family visits, and maintaining required distances at home. The recommended COVID-19 prevention protocol and rules also highlighted the necessity to avoid certain conventional social behaviors such as shaking hands, embracing, and kissing [10,11]. In addition, the rules prohibited older adults from going to public places, shopping centers, worship places, and other public gatherings [12]. The UAE government also warned about social distancing through various communication channels, i.e., broadcasts, TV announcements, newspapers, and social media. Acknowledging that social distancing may aggravate loneliness among older people, the messages had a positive and promising tone that conforming to such rules will increase the chances to end the pandemic sooner [13].

In the Emirate of Abu Dhabi, as part of its efforts to improve the health level of older adults, the Family Development Foundation (FDF) implemented several initiatives to control and minimize the spread of COVID-19 [14]. First, in cooperation with the Department of Health, the FDF focused on campaigns of ‘Educating Senior Citizens and Residents’ in various regions of Abu Dhabi, with the ultimate goal to ‘increase their health awareness levels and enable them to adopt the preventive precautionary measures that contribute to maintaining their health and safety.’ As a step further, family members, friends, local charities, voluntary organizations, and community organizations were encouraged to develop comprehensive networks to ensure each older person has some meaningful social contact to support [15]. Through such organized and comprehensive approaches, social charities, organizations, and healthcare providers could work together to support older people through this period of social isolation and loneliness. The government also communicated with older adults in Abu Dhabi to inform them that they were not required to go to testing centers for testing for COVID-19 [16]. Instead, related government units called them and recommended a public health representative to visit them at home for the testing. All services for older adults were provided free of charge.

The role of technology in combating the pandemic was apparent in Abu Dhabi. The public messages sent by concerned government agencies stressed the influential roles of technology in keeping communications active, especially concerning older adults [17]. During this social crisis, older adults were encouraged to take advantage of those communication technologies to think about unique opportunities for individual learning, career development, and communicating with family and friends [13].

This research focused on older adults (60+ years old) in the Emirate of Abu Dhabi of the United Arab Emirates (UAE) during the COVID-19 pandemic. The COVID-19 countermeasures in Abu Dhabi started to take shape since the pandemic and first reported cases, namely since March of 2020 [17,18,19,20,21]. Due to vast restrictions imposed on the public and, more specifically, on older adults, it was indeed significant to better understand the implications. The study’s primary purpose was to investigate specific psychological and social concerns and challenges reported by older adults. The research focused on the most common psychological feelings reported in other international studies. The psychological feelings included fear, loneliness, sadness, irritability, emotional exhaustion, depressive symptoms, sleeping disorders, overeating, and excessive screen use. In addition, the objectives included understanding the effect of new technology and the trend influence of changes imposed, more specifically, the weight of older people. A descriptive approach was adopted, drawing on the Abu Dhabi COVID-19 survey results. This research also aimed to test the effect of time on certain feelings and challenges amongst older adults. It was hoped that the results of this research would provide valuable insights for public social policymakers to understand better the actions they need to take in such crucial circumstances. 

The objectives of the study could be expanded into more detailed secondary objectives. They include understanding the descriptive nature of the ten psychological feelings felt most by older adults (mainly using descriptive means and standard deviations of the psychological feelings); understanding the changes in the ten psychological feelings according to time (mainly using trend analysis of the ten psychological feelings as dependent variables, where time was treated as the only independent variable); identifying the most severe challenges that elderlies feel during the pandemic (mainly using descriptive means and standard deviations of the challenges faced by older people); understanding the pandemic effects on the bodyweight of older adults (mainly percentages of older adults scoring on each of the five possibilities concerning weight loss); understanding the effect of staying at home during the pandemic using new technologies (mainly means and standard deviations of each of the practices); and understanding the differences in mean reactions between the sample of older adults (+60 years old) and the rest (all age categories).

## 2. Literature Reviews

Older adults are at a significantly increased risk of severe disease following infection from COVID-19. The WHO announced in April 2020 that more than 95% of COVID-19 deaths were among people over 60 years of age, and more than half of all deaths occurred in people of 80 years plus [18]. In Sweden, for example, 90% of the deaths from COVID-19 were among people more than 70 years of age [19]. Furthermore, there have been distressing international news and reports of older people abandoned in care homes during the pandemic [20,21]. In addition, numerous media coverage and online commentary about the potential rationing of care with older people suggest that older adults are potentially more disadvantaged than persons in other age groups [22,23].

The COVID-19 pandemic literature, in general, has reflected concerns related to common health issues and mental health risks associated with older adults. Apart from some physical health issues such as weight loss [24,25], the literature has recorded a wide range of psychological feelings, challenges, and concerns reported by people during the pandemic. These psychological feelings and challenges include untold fear and suffering [3], sadness [26], loneliness [6,26], stress [27,28], irritability [29] (Patel, 2021), emotional exhaustion [30], depression [26], sleeping disorders [31], overeating [32], and excessive screen use [33]. Research has also concentrated on various concerns and challenges affecting the whole community in general and older adults, including imposed restrictions [6,26], not being able to go out in public [34], disturbance of social life [27], less physical activities [18], less access to regular medical visits [35], less get-togethers with younger children [34], and loneliness [6,26].

Social isolation has been identified as a severe public health concern among older people [36,37]. As a subjective and complex emotion, loneliness reflects a lack of contact with or physical separation from family and friends. The isolation might be broadened to include social networks and the lack of involvement in social activities. Some [38] have stressed that social isolation is usually experienced as a feeling of anxiety and dissatisfaction associated with a ‘lack of connectedness or communality with others, and a deficit between the actual and desired quality and quantity of social engagement.’ It is worth noticing that social isolation and loneliness are correlated and often used interchangeably [39]. Most studies have acknowledged that older people’s social isolation and loneliness are essential and paramount due to the detrimental and influential impact on their mental and physical health [40]. Loneliness is a real risk factor to the health and well-being of all people, where older people can be more vulnerable to being lonely [26]. Social isolation and loneliness increase older people’s risk of anxiety, depression, cognitive dysfunction, heart disease, and mortality [41]. Social isolation might lead to other outcomes, affecting older adults’ level of awareness and knowledge.

In addressing the challenges faced by older people, some [42] have elaborated that we need to examine both the physical and social impact and the underlying reasons. Empirical research has shown that COVID-19 negatively impacted older adults’ body weight and nutritional status [24,25,42], suggesting a higher risk for lockdown-induced weight loss for the older population. The results of the Canadian Perspective Survey indicated a significant impact of COVID-19 on screen time and mental health [43], as more than 60% of respondents reported increasing TV time and internet usage. Similarly, [44] reported that the COVID-19 pandemic had increased people’s screen time for various reasons, including increased time spent on virtual education, working from home, online shopping, and electronic communication with friends and family. The positive association between family members’ older adults’ wellbeing has been studied in many settings [45]. Some [46] used logistic regression to determine how family size affects psycho-social, economic, and health wellbeing in old age in Mexico. Their study showed that having fewer children is associated with a lower economic wellbeing and higher odds of being uninsured for the older cohort. Older adults with children are more likely than older adults without children to have frequent social interactions. Social contacts such as this offer emotional and instrumental support that enhances wellbeing throughout the life course, and the importance of these contacts is especially evident at advanced age levels [47].

In terms of public policies and programs, health and social care policies and campaigns worldwide acknowledged the issue of loneliness and social isolation well before COVID-19. For example, the Campaign to End Loneliness in the UK helped create a vast network of national, regional, and local organizations to work cohesively and ensure that the social isolation and loneliness of older people remain a public health priority [48]. In addition, the New Zealand government has emphasized its commitment to an aggressive vision of positive aging principles to promote community participation and prevent social isolation and loneliness [49]. 

It should be realized that in most developed countries, a different population of older people at risk of becoming more socially isolated resides in residential care homes. During the COVID-19 pandemic, such elderlies encountered challenges as their family members and friends were not permitted to visit them, although most countries adopted regular consultation with medical and related professionals to support older people [50]. Therefore, there is an urgent need to support older people as there might be more negative impacts on their physical and mental health from social isolation and ageist discourses around COVID-19. In such conditions, older adults might require more support to have and retain their connectedness and communality to better enable a sense of belonging. In the United Kingdom (UK), an example of support was rendered by Public Health England, which issued a publication of guidance on maintaining mental health and well-being during COVID-19 social restrictions [51]. The UK government also embarked on a media campaign to recruit volunteers to support older people who needed assistance. The campaign was taken as an inclusive approach to ensure that older people are not left isolated over an extended period. Other responses to socially support older people included the development of social networks through online technologies. 

The adverse effects of COVID-19 are more severe and compounded for older adults who do not have access to modern technology platforms or could not acquire more effective means of communication [52]. Empirical research has focused on the technical aspects of the pandemic related to older adults [6,50]. Several studies have promoted online technologies to provide social support networks and a sense of belonging during the pandemic [53]. Some have also suggested that cognitive-behavioral and psychological therapies be delivered online to decrease loneliness and improve mental wellbeing [6]. Other creative ways of supporting the general population, including older people, included more creative online applications and online platforms [54]. However, such technological aiding facilities might be inhibited by disparities among the older population in the access to or literacy in digital resources [50]. As a result, some analysts recommended that interventions involve more frequent communications via telephone, messaging, SMS, or other simple social media resources [55,56,57].

## 3. Methods and Design

### 3.1. The Survey Instruments

The Abu Dhabi Community Development Department (ADDCD) developed the survey instrument in cooperation with the Statistics Department Abu Dhabi (SCAD). The relevant literature provided the basis for the essential dimensions in the survey to be developed, mainly threats and challenges that elderlies might experience during rough times such as COVID-19. Several international surveys were consulted [58,59,60]. 

The original instrument included a multitude of dimensions such as time-related expectations, COVID-19 awareness and attitude, economic and business challenges, education-related concerns, health concerns, mental health concerns, community and daily practices, technology-related challenges, and concerns about and trust in government responses. The survey was modified as time went by to include relevant variables according to the pandemic developments. This current study selected the following elements from the survey relevant to the theme of the study. The survey asked respondents to rate on a scale of 5 the extent of some psychological feelings being developed since the outbreak of the COVID-19 pandemic: fear, loneliness, sadness, stress, irritability, emotional exhaustion, depressive symptoms, sleeping disorders, overeating, and excessive screen use. The survey also asked older adults to rate on a scale of 5 the degree of eight specific challenges. They included more restrictions imposed, not being able to go out to public places, social life disturbed more than before, lack of physical activity, lack of access to regular medicine/physiotherapy, not having necessities like food, not seeing grandchildren whenever desired, and being lonely. In addition, the survey asked older adults to portray their level of agreement on a scale of five with the role of technology. Four items were included: ‘new technologies contribute to a better quality of life’; ‘I can usually figure out new high-tech products and services without help from others’; ‘sometimes technology systems are not designed for use by ordinary people’; and ‘technology lowers the quality of relationships by reducing personal interactions.’ The survey also asked older adults to rate their level of interest in seeing their primary care physician via a virtual visit, from 1 (not at all interested) to 5 (very interested). For those who selected ‘very interested’ or ‘somewhat interested,’ a list of reasons was provided to understand better why they were interested in seeing their physicians online. Finally, a question asked respondents about their weight status if the pandemic had any impact. The options included a 5-point Likert scale from underweight to overweight. A further question was asked to identify the number of kilograms gained or lost for overweight or underweight. The survey was administered from February to September 2020. With regard to the demographics of respondents, the survey collected various information concerning age, gender, level of education, residential location and region, type of residents, monthly household income, nationality, marital status, number and type of family members, type and category of work, and health characteristics.

### 3.2. Study Sample and Survey Distribution

The study sample included residents across the three regions of Abu Dhabi: Abu Dhabi region, Al Ain region, and Al Dhafra region. The survey team made extra efforts to reach all community residents to achieve representative samples. The survey acquired more than 33,000 responses, among which older adults (60+ years old) accounted for 578 respondents. The survey was available in Arabic, English, and six other Asian languages. The survey was distributed online. More than 50 survey links were created and distributed amongst the various segments of the community. Both ADDCD and SCAD were involved in distributing the survey links. ADDCD also sent encouraging calls to the communities, inviting their participation in the survey. The survey was distributed in two stages, at the start of the COVID-19 pandemic and after four months. Means of distribution included phone calls, messengers, emails, and social media. Survey representatives also appeared in several national TV newscasts to encourage participation. It should be added here that the online means of distribution facilitated reaching respondents who were not in the country at the time of distribution. 

### 3.3. Analysis Methods 

The analysis mainly used descriptive analysis, *t*-tests, analysis of variance (ANOVA), and simple trend analysis. To understand respondents’ psychological feelings, concerns during the pandemic, and views on the effect of new technologies, we presented the means, standard deviations, and one-sample *t*-tests for the older adults compared to other age groups. The two-staged survey distribution allowed us to capture the effect of more restrictions imposed on older adults in Abu Dhabi in the post-COVID-19 era. We utilized simple regression for the trend in psychological feelings by recording the *t*-values and their associated significance levels. To understand the pandemic concerns, we utilized simple descriptive statistics of means and standard deviations and the appropriate one-sample *t*-tests. In addition, we presented the percentages of the mean changes in weights of respondents. The means of the overall level of agreement with the role of technology was also addressed along with the one-sample *t*-tests. For the psychological feelings, challenges and concerns, and the reactions regarding new technologies, further analysis (one-sample *t*-tests) explored the mean differences between the older adults’ sample and the whole sample of respondents in all age categories. To understand differences according to gender, marital status, and education attainment, we performed analysis of variance (ANOVA). The software (SPSS-27) was used throughout the analysis [61].

## 4. Results

Table 1 shows the breakdown regarding specific categories. About 60% were male and 40% were female. Most of them were married (82.4%), while only (7.7%) were single. Around (3.1%) were separated, widowed, or divorced. About 40% were Emirati and 60% were non-Emiratis. Regarding education attainment, the most significant percentage (39.4%) of the elderlies held a bachelor’s degree, while 3.7% held doctorate degrees. Those not holding any degrees below bachelor’s degrees accounted for (47.5%). Regarding housing, more than 42.8% lived in a villa, 46.9% in an apartment, 2.9% in collective housing, and 7.4% in other forms of housing. It should be realized that the percentages of the different categories reflect the accurate representations of each Abu Dhabi. However, regarding nationality, Emiratis in Abu Dhabi are less than the 40% that is reflected in the response rates. Therefore, specific weighting was used to represent the actual percentages for this category. No differences were observed compared to unweighted results. We should reflect here that both Emiratis and non-Emiratis experienced the same healthcare attention from the public institutions in the country.

Most of the older adults were recorded between 60 and 67 years old (64.6%). Respondents also noted that (9.36%) of them lived alone, while 90.64% had at least one person living with them. 

***Psychological feelings***—The results in Table 2 show the means and standard deviations regarding self-reported psychological feelings during the pandemic. The mean values for these psychological health attributes were relatively low (below 3.0), with the highest mean of 3.2314 for excessive screen use, 2.6552 for fear, 2.6534 for loneliness, and 2.6063 for stress. On the other hand, the feelings of depressive symptoms and overeating scored among the lowest, at 1.9203 and 1.9059, respectively. Further ANOVA revealed no differences regarding gender and marital status. 

However, significant differences were observed for educational attainment regarding five of the psychological feelings (loneliness, irritability, emotional exhaustion, sadness, and overeating), where those below college degrees recorded the highest means. A one-sample *t*-test (comparing the means to the overall population mean of all ages) resulted in the significance of all psychological feelings except (loneliness). The negative *t*-values indicate that older adults provided (smaller) means for the corresponding feelings compared to the rest of the population sample in the survey.

***Trends in psychological feelings***—The analysis took each of the ten psychological feelings as dependent variables, where time was treated as the only independent variable. Table 3 shows the parameters that resulted in ten individual regression analyses. Only two variables, sleeping disorders and overeating, showed significance at the 0.05 level. The positive standardized coefficients and *t*-values note more negative development for the two feelings for older adults, suggesting that time harms the development of the two feelings. The same analysis for those below 60 years shows significant differences concerning four variables—stress, irritability, emotional exhaustion, and excessive screen use.

***Pandemic concerns***—Table 4 shows the most severe challenges that elderlies feel during the pandemic. Again, three variables scored a value above 3.0. Regarding the most severe challenges that elderlies are experiencing during the pandemic, the variable ‘more restrictions imposed’ on elderlies stood alone with the highest level of concern (3.462), followed by ‘unable to see children and grandchildren whenever desired or wanted’ (3.124) and ‘not being allowed in public places’ (3.056). 

On the contrary, the three variables that received the lowest concern were ‘lack of access to regular medicine/physiotherapy’, ‘being lonely’, and ‘not having necessities like food’. About 29.24% of older adults indicated that they are interested in seeing their primary care physician via video during and after COVID-19. This question also asked respondents to elaborate further on the reasons for seeing their physician. The four main reasons elderlies are interested in seeing their physician via virtual visit after COVID-19 were prescription renewals, addressing a common illness (i.e., cold, flu), managing chronic illness, and psychological consultation. 

A one-sample *t*-test resulted in the significance of five of the challenges. They included (being lonely, not having necessities like food, lack of access to medicine, lack of physical activity, and more restrictions imposed). The nonsignificant concerns indicate more similarity between the older adults’ challenges and those of other sample categories. Further ANOVA of the pandemic concerns revealed no gender and marital status differences. However, significant differences were observed for educational attainment regarding the significant concerns or challenges (loneliness, emotional exhaustion, and sadness). Again, those below college degrees recorded the highest means.

***Other pandemic effects and reactions***—The results, in general, do not indicate that the pandemic had a significant effect on the body weight of older adults. Only 2.47% felt that they gained too much weight. About 20.85% felt that they gained weight. The majority (65.37%) felt that they maintained their weight. Meanwhile, 9.47% perceived losing weight, and 1.77% said they lost too much. For older adults and those who reported a gain in weight, the mean of gained weight was 4.32 kg. For those who reported weight loss, the mean of lost weight was 5.52 kg (Table 5). Further ANOVA of other pandemic effects and reactions on weights revealed no gender, marital status, or educational attainment. 

***Communication and technology***—For staying at home during the pandemic, it was necessary to explore further the perception and readiness of elderlies to deal with new technology. The mean scores for the four statements concerning the role of technology are shown in Table 6. The highest mean was assigned to ‘new technology contributes to a better quality of life,’ as older adults recorded a mean of 3.866. The statement ‘I can figure out new high-tech products and services without help from others’ recorded a mean of 3.543. The statement ‘sometimes new technologies are not designed for use by ordinary people’ scored a mean of 3.665. Finally, ‘technology lowers the quality of relationships by reducing personal interactions’ scored a mean of 3.795. In general, all four means were above the middle point of 3.0. 

One-sample *t*-test results of the significance of the four new technology-related variables produced significance regarding two of the variables (new technologies contribute to a better quality of life and figure out new high-tech products without help from others). For both variables, older adults reported smaller mean values. The further ANOVA of the pandemic concerns revealed no gender and marital status differences. For educational attainment, significant differences were observed regarding the psychological feelings of (loneliness, emotional exhaustion, and sadness). Those below college degrees recorded the highest means.

## 5. Discussion

The study investigated significant developments concerning older adults during the pandemic. It explored their perception regarding ten psychological feelings. Moreover, it examined if the feelings changed during the pandemic. The analysis also investigated older adults’ concerns and challenges faced during the pandemic. In addition, the investigation looked closer at their perception of health and related issues concerning losing or gaining weight and opportunities for seeing their medical advisors. Finally, it presented older adults’ opinions about the role and effect of technology while staying at home. In many survey questions, older adults provided mean responses, which were lower than the total respondents who participated in the survey. Depending on the nature of the question, most older adults provided means lower in magnitude than those of other age brackets in the population sample.

Relevant research in other countries supports the presence of certain psychological feelings with older adults in Abu Dhabi. The result of excessive screen time during the pandemic for older adults in Abu Dhabi is consistent with the results reported in several studies [33,44]. Among these findings, the issue of loneliness as a significant feeling for older adults has taken center-stage attention in older adults’ related research around the world, for which the present Abu Dhabi study provides further support. Our results consistently suggest that loneliness is a real risk factor to the health and well-being of all people, especially older people who could be more vulnerable to being lonely [51,62]. In addition, the reported feeling of stress is evident in the Abu Dhabi study, which is in line with the empirical findings [27,28]. Overall, the results from Abu Dhabi echo other international research that shows that the COVID-19 pandemic has been a significant stressor that has affected older adults worldwide [30].

However, we should also note that the perceived level of various psychological feelings or disorders associated with the COVID-19 pandemic tends to be low among Abu Dhabi’s older adults. Some psychological feelings such as irritability, emotional exhaustion, depressive symptoms, sleeping disorders, and overeating received relatively low means. Such results are not consistent with similar surveys of elderlies in other communities worldwide [29,30]. It is worth emphasizing that, on the one hand, Abu Dhabi and the UAE, in general, have government institutions and specific policies dealing with the wellbeing of older adults. For example, the government of Abu Dhabi issued many policies that centered on older adults and the issue of social connection [63]. On the other hand, the traditionally extensive and strong social connection has remained in the modern life of Abu Dhabi, where older adults usually live at home with other family members. Inside the family, they are allowed to practice minimum isolation procedures during the pandemic (i.e., reduced visits from family members and social distancing among family members). In this case, the circumstances in other countries may be relatively different from Abu Dhabi [6,26,64]. When we focus on the psychological feelings, pandemic concerns, effects on weights, and effect of technology, the results indicate no significant differences in gender and marital status in all related variables. However, some more negative outcome variables were significantly different for those holding below-college degrees. 

In this context, the UAE is known for nursing care at home or Home Health Care (HHC) rather than nursing homes. Home care has multiple benefits, from companionship to promoting aging with dignity [65]. Home care has also carefully selected nurses and professionals who are passionate and well-trained. Indeed, HHC foraging has become the preferred solution for the older adult population. When provided the opportunity, most senior citizens prefer to live out their later years in the comfort and familiarity of their homes. With the help of licensed medical professionals, they can avoid moving to an unfamiliar assisted living facility. There are relatively few older adult care facilities in the UAE, where the norm remains that caring for older adults is the duty of family members. 

For Abu Dhabi’s older adults, the ten perceived psychological feelings experienced relatively minor changes during and post-COVID-19. The few months had some adverse effects only on sleeping disorders and overeating. It seems that spending more time at home produced some restrictions and plenty of spare time. Research in other countries also produced such effects [31]. However, it should be noted that most research concerning sleeping or overeating covered the general population [3,66].

Regarding COVID-19-related challenges, it was clear that older adults saw more restrictions imposed on them to be more of a challenge than other age categories. This specific result entails the feelings of isolation noted by [67]. Social isolation might be due to environmental and policy restrictions rather than an individual’s ability to create or maintain social relationships [6,26]. The COVID-19-related restrictions imposed by the Abu Dhabi government also included fewer or restricted visits of family members, which explains the second-highest rated challenge by older adults, that is, ‘unable to see children and grandchildren whenever desired or wanted.’ Such a concern was referred to in many papers [67]. Furthermore, elderlies in Abu Dhabi also ranked ‘not being allowed to public places’ as a significant concern. Similar results have also been reported in other countries such as Japan [34]. However, those three low-rated challenges or concerns, i.e., lack of access to regular medicine/physiotherapy, loneliness, and lack of necessities such as food and medicine, are not consistent with the research findings in other countries [35]. 

The results of this study pointed out that older adults in Abu Dhabi felt comfortable figuring out new high-tech products and services without help from others. They also tended to have a more favorable attitude toward virtual technology and appreciate the role of new technology in contributing to a better quality of life, which is consistent with research findings in other countries [21]. However, Abu Dhabi’s older adults realized that, sometimes, new technologies are not designed for use by people with low technology literacy. Moreover, they pointed to the adverse effects of technology in lowering the quality of relationships by reducing personal interactions. Such results support the efforts of public institutions in Abu Dhabi and elsewhere [20]. In Abu Dhabi, public messages sent to communities by government authorities encouraged all family members, especially older adults, to upload mobile applications that facilitate visual communications online. They also offered tutorials for educating older adults. 

This research explored people’s understanding of the effect of the COVID-19 pandemic on their preference to shift in health care delivery platforms, necessitating a new reliance on technology and telemedicine. The results indicated that a relatively low percentage of older adults in Abu Dhabi reported a positive response of their interest to see their primary care physician via a virtual visit during and after the COVID-19 pandemic, much lower than some reported figures of older adults that completed or scheduled telemedicine visits in other parts of the world [68,69,70,71,72,73,74]. This outcome may be explained by the privileges that the government of Abu Dhabi has given to older adults during the pandemic, as government medical facilities were encouraged to call elderlies to request home visits [12], and all services to elderlies were provided free of charge.

The research included specific symptoms such as fear, loneliness, sadness, stress, irritability, emotional exhaustion, depressive symptoms, sleeping disorders, and overeating. However, one limitation in this study has to do with the fact that each symptom was asked by one question. Hence, one might doubt if respondents could accurately reflect the situation. Another limitation is that the socio-demographic characteristics of participants were not covered fully regarding the psychological feelings. We should also refer to some potential limitations derived from the sample collection that could be avoided in the future. First, the survey depended mainly on online distribution. Future similar surveys could solicit older adults’ responses by conducting home visitations and interviews for those with reservations regarding online technology. Several months after this specific survey, vaccination started in the UAE. Future research might address how the availability of vaccination might have changed the results.

## 6. Conclusions

This research acknowledges that many papers have already discussed older adults’ mental health issues during the COVID-19 lockdown. However, we should stress that the Abu Dhabi community has unique features. Older adults in Abu Dhabi live within a unique culture where they have special uniqueness regarding tradition, respect, and social connection. This paper adds many new pieces of knowledge in this area. 

Regarding the first objective, the overall self-reported psychological feelings during the pandemic recorded relatively low values (below 3.0 out of 5). Excessive screen use recorded the highest mean. The psychological feelings of fear, loneliness, and stress recorded the highest means. On the other hand, feelings such as depressive symptoms and overeating scored lowest. For the second objective, the trend analysis of ten psychological feelings with time recorded significant negative changes for only two of the psychological feelings, sleeping disorders and overeating. The third objective dealt with specific challenges encountered by older adults. The highest challenges were more restrictions imposed on elderlies, unable to see children and grandchildren whenever desired or wanted, and not being allowed in public places. The challenges with the lowest concerns were the lack of access to regular medicine/physiotherapy, loneliness, and lack of necessities like food. The fourth objective dealt with the effect of the pandemic on the body weight of older adults. The results showed that only a quarter of older adults’ recorded gaining weight, as most maintained their weight. The final objective was to explore further the perception and readiness of elderlies to deal with new technology. The older adults assigned a high importance score, describing new technology to contribute to a better quality of life. However, they recorded a severe concern regarding their need to receive more help and assistance regarding new high-tech products and services. They saw it as ‘not designed for use by ordinary people’, and the worry that ‘technology lowers the quality of relationships by reducing personal interactions’.

The Abu Dhabi study provides further support that loneliness is a real risk factor to the health and wellbeing of older people who are more vulnerable to being lonely. Overall, the results echo other international research that shows that the COVID-19 pandemic has been a significant stressor that has affected older adults. Contrary to results in other countries, some psychological feelings such as irritability, emotional exhaustion, depressive symptoms, sleeping disorders, and overeating did not show severe concerns in Abu Dhabi. In this regard, it is worth emphasizing that government policies dealing with the wellbeing of older adults provided some positive reactions. The traditionally extensive and strong social connection has remained a primary supporting feature for families living in Abu Dhabi as cultural aspects of closeness have enriched the opportunity for older adults to still be close to their loved ones at home. Overall, the results showed no significant differences in gender and marital status in all related variables. Overall, the feelings of isolation for older adults were enhanced by the more restrictions imposed on them. As a result, their inability to see their children and grandchildren was a significant challenge. The results revealed that older adults witnessed a more favorable attitude toward virtual technology as they appreciate its roles in contributing to a better quality of life. However, they realized that sometimes new technologies are not designed for use, as they might not be prepared enough to deal with it, as it lowers the quality of relationships by reducing personal interactions. However, they were encouraged by related government authorities’ support with mobile applications that facilitate visual communications online. They also offered tutorials for educating older adults. 

In a relative sense, older adults in Abu Dhabi rated some specific psychological health attributes as more challenging, including excessive screen use, fear, loneliness, and stress. They rated the feelings of depressive symptoms and overeating relatively low. Trend analysis showed significant changes in sleeping disorders and overeating and pointed to improvements in other feelings but not significantly. Older adults revealed several concerns associated with the pandemic, such as more restrictions imposed on elderlies, not seeing children and grandchildren whenever desired or wanted, and not being allowed to go to public places. On the other hand, they did not show much concern concerning the lack of access to regular medicine/physiotherapy and not having necessities.

During difficult times such as the COVID-19 pandemic, a powerful family bond can make all the difference in the world for older adults. Families provide a stable foundation for emotional support for their loved ones. The home environment is the least restrictive place for older adults to remain engaged with their typical daily activities in the community and with family and friends around them. Abu Dhabi government institutions could further design volunteer schemes to better communicate with older adults. For example, special volunteers could have home visits to educate older people to use IT and virtual applications. Such initiatives might increase their health awareness levels and enrich their pandemic knowledge. In addition, such home visit projects by volunteers or medical teams could act as ambassadors to educate older adults and senior citizens about coronavirus-related preventive measures.

Future research should investigate if different people had a different definition of these symptoms. There is a need to understand better the existence of significant differences in gender, age, education status, income status, marital status, nationality, and other relevant characteristics. More analysis is needed to provide elaborations on the meanings or definitions of each term to guarantee they meant the same thing. Knowing baseline characteristics of the elderly respondent, their comorbidities, and pre-pandemic psychological conditions could help further understand the specific impact of the pandemic on mental health. Future research should also investigate the effect of the number of people and older adults’ quality of life in more comprehensive ways. It should also be said that multiple regression analysis is only shown for psychological feelings. More analysis could investigate the association between physiological concerns and other variables (i.e., demographic). Such results could provide more relevant information to enrich policymaking accordingly. A further objective should try to revisit the issues raised by this study to confirm the results and continue exploring the effects of family settings. When it comes to times of more complex circumstances such as pandemics, being consistent with other international research, it might be more informed to include other related variables such as smoking, sport and activities, specific social behaviors, and economic-related variables in order to generate a more comprehensive picture of older adults’ wellbeing. In addition, a comparison between Abu Dhabi and another aging-developing country may be helpful for better policy makings. 

## Figures and Tables

**Table 1 ijerph-18-13427-t001:** Respondent’s profile.

Gender	Percentage
Male	60.1%
Female	39.9%
Marital status	
Married	82.4%
Single	7.7%
Divorced	1.5%
Separated	0.8%
Widowed	0.8%
Education level	
Illiterate	1.3%
Below secondary school	4.2%
Secondary school	15.6%
Post-high school training certificate	17.7%
College diploma	8.8%
Bachelor’s degree	39.4%
Master’s degree	8.3%
Doctorate degree	3.7%
Age	
60–63	33.0%
64–67	30.6%
68–71	21.3%
72 and older	15.1%
Nationality	
Emirati	40.3%
Non-Emirati	59.7%
Housing type	
Villa	42.8%
Apartment	46.9%
Collective housing	2.9%
Other types	7.4%
Live alone or with others?	
I live alone	9.36%
Share home with others	90.64%

**Table 2 ijerph-18-13427-t002:** Psychological feelings during the pandemic.

Psychological Feelings	Mean	Rank	Standard Deviation	*t*-Value	Sig.
Fear	2.6552	2	1.165	−4.068	0.001
Loneliness	2.6534	3	1.301	−0.985	0.325
Sadness	2.4960	5	1.230	−4.695	0.001
Stress	2.6063	4	1.252	−7.400	0.001
Irritability	2.2941	7	1.201	−8.434	0.001
Emotional exhaustion	2.3571	6	1.261	−7.188	0.001
Depressive symptoms	1.9203	9	1.203	−7.754	0.001
Sleeping disorder	2.0745	8	1.293	−7.663	0.001
Overeating	1.9059	10	1.150	−10.297	0.001
Excessive screen use	3.2314	1	1.355	−6.777	0.001

**Table 3 ijerph-18-13427-t003:** Regression analyses of perception of the ten psychological feelings.

Model	Unstandardized Coefficients		Standardized Coefficients	*t*-Value	Sig.
	B	Std. Error	Beta		
Fear	0.012	0.011	0.055	1.057	0.291
Loneliness	0.004	0.013	0.018	0.335	0.738
Sadness	−0.010	0.012	−0.045	−0.845	0.398
Stress	0.014	0.012	0.060	1.137	0.256
Irritability	0.013	0.012	0.060	1.132	0.258
Emotional exhaustion	0.014	0.012	0.059	1.121	0.263
Depressive symptoms	0.006	0.012	0.028	0.524	0.601
Sleeping disorder	0.027	0.012	0.114	2.181	0.030
Overeating	0.025	0.012	0.114	2.177	0.030
Excessive screen use	0.019	0.013	0.075	1.412	0.159

**Table 4 ijerph-18-13427-t004:** Older adults’ major concerns and challenges.

Challenges	Means	Ranks	Standard Deviations	*t*-Value	Sig.
Being lonely	2.427	7	1.207	−17.301	0.001
Not being able to see my grandchildren whenever I want	3.124	2	1.323	0.78	0.938
Not having necessities like food	2.555	6	1.403	−6.602	0.001
Lack of access to regular medicine/physiotherapy	2.315	8	1.303	−4.579	0.001
Lack of physical activity	2.654	5	1.261	4.332	0.001
Social life disturbed more than before	2.921	4	1.194	−0.768	0.443
Not being able to see my grandchildren whenever I want	3.124	2	1.323	−0.326	0.744
More restrictions imposed	3.462	1	1.287	15.604	0.001

**Table 5 ijerph-18-13427-t005:** Overall weight (changes).

Weight Changes	Percentage
Lost too much weight	1.77%
Lost some weight	9.47%
Maintained my weight	65.37%
Gained weight	20.85%
Gained much weight	2.47%

**Table 6 ijerph-18-13427-t006:** Overall level of agreement with the role of technology.

Technology	Mean	*t*-Value	Sig.
New technologies contribute to a better quality of life	3.866	−2.529	0.043
I can usually figure out new high-tech products and services without help from others	3.543	−5.996	0.001
Sometimes technology systems are not designed for use by ordinary people	3.665	0.311	0.756
Technology lowers the quality of relationships by reducing personal interactions	3.795	0.445	0.657

## Data Availability

The data presented in this study are available on request from the corresponding author. The data are not publicly available, due to privacy restrictions.

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
