# Peer review of "Experiencing the Unprecedented COVID-19 Lockdown: Abu Dhabi Older Adults’ Challenges and Concerns"

_ijerph, 2021, doi:10.3390/ijerph182413427_

Round 1

Reviewer 1 Report

The manuscript ijerph-1487279-entitled “Experiencing the Unprecedented COVID-19 Lockdown: Abu 2 Dhabi Older Adults’ Challenges and Concerns” have addressed most of the comments and suggestions recommended in the previous submission. The quality of the article has been significantly improved. The manuscript has incorporated appropriate explanations to support the reported findings. Statistical analysis of the studied variables has been now taken into account. References have been sufficiently updated and the presentation of Tables and Figures has also improved.

Several aspects still need to be considered before publication:

  • Abstract: The inclusion of p values for the domains that have shown statistical significance could be useful to demonstrate the relevance of the study. The result of the model regression analysis could also provide appropriate information.

  • Methods: The subsection Study objectives should not be included in this section and, as mentioned before.

  • Discussion: potential limitations derived from the sample collection could also be considered.

  • Several typo errors should be revised.

Reviewer 2 Report

The authors have carried out the recommended improvements in terms of broadening the main objective, as well as formulating secondary objectives so that they describe in a more detailed way the variables to be analyzed.

On the other hand, the Discussion section has also been expanded, where more contributions of the results have been included.

Also, the information in the conclusions section has been reorganized, connecting each objective with its conclusion.

Finally, more practical applications have been added and put after future research, which has resulted in the quality of work.

Reviewer 3 Report

Please provide additional information about the specific development of the pandemic in your country (when did it start? at which time did the government start to postpone lockdowns?) 

Your cohort contains a high amount of non-Emiratis, can you give further information? Where do they live? How could local differences in health care systems influence the results? If the survey was performed online, was it possible to participate from other countries around the world? 

Do you have information on how availability of vaccination changed the results? 

Language and style 

Please provide additional language editing and spell check before resubmission. 

Overall interesting data, unfortunately lacking a control group. Could be imporved by adding data of younger patients in comparison 

Round 2

Reviewer 3 Report

No further concerns after providing the required information. 

Author Response

The reviewer added (English language and style are fine/minor spell check required). We checked the paper for style and spell check). The whole paper is run through (Grammarly). 

The conclusion is also expanded and enhanced to support the results. 

This manuscript is a resubmission of an earlier submission. The following is a list of the peer review reports and author responses from that submission.

Round 1

Reviewer 1 Report

It is recommended to expand the main objective, as well as formulate secondary objectives so that they describe in a more detailed way the variables to be analyzed.

Regarding the instruments, it is recommended to include the name of the main instrument as well as its Cronbach's Alpha.

In the discussion of the work, the contributions of the results should be further expanded, since they do not stand out enough, giving a greater role to the studies that support them.

In the conclusions section, it is recommended to reorganize the information, as well as expand it, connecting each objective with its conclusion.

Expand practical applications and put them after future research.

Review the description of future limitations because “Future research” is repeated several times in the same paragraph.

Reviewer 2 Report

The manuscript  ijerph-1430916 “Experiencing the Unprecedented COVID-19 Lockdown: Abu 2 Dhabi Older Adults’ Challenges and Concerns” investigates the impact of the COVID-19 pandemic on the feelings and concerns of 574 older adults from Abu Dhabi.

This work presents an interesting approach for studying the effect of the restrictions derived from the pandemic from April 11 to July of 2020. The topic is relevant and the authors were in a good position to conduct the analysis. The contents are interesting and can be useful for the journal audience. However, the manuscript lacks clarity and consistent data in its present form. The presentation of experimental results could be improved and the discussion section needs to be improved. For these reasons, it is recommended the exhaustive revision of the present version before publication.           

I would also like to comment on the following issues that the authors should consider:

- Abstract: the organization of information needs to be adjusted. A short introduction or background should be mentioned. Specific data regarding the sociodemographic characteristics should be placed after the objective. The type of study design, as well as the details of the questionnaire, should be considered. The statistical significance of some results such as the opinions and change in body weight should be included.

- Introduction: the text should be reorganized to present the information in a more ordered manner. The aim of the study as well as the study’s main contribution is commonly placed in the final part of the introduction. The literature review or background should be located following the presentation of the topic. Additionally, the extant knowledge on the evaluation of psychological and socioemotional during the pandemic should be included, rather than the type of measures implemented by governments around the world.  A broad number of works have addressed the mental health effects of COVID-19 in older adults from last year until now. The references should be adequately updated to incorporate recent advances in this context. Regarding the overview of COVID-19 counter-measures in the UAE and Abu Dhabi, it should be stated if these measures were implemented before or after the development of the study.

Methods: This section should specifically report the questions included in the survey. Socio-demographical characteristics of the participants should be represented in a Table in the Results section. The data analysis should also mention the type of statistics used as well as the software employed to perform the analysis.

- Results and discussion: the following aspects should be considered.

  • As mentioned above, the demographics and characteristics of the study population should be specifically represented in a Table. The age range of participants is not mentioned, only the inferior limit.
  • The data displayed in Table 1, Table 3, and Table 4 include neither t-test nor statistical significance values.
  • Data referring to bodyweight could be presented in a Table.
  • Baseline characteristics of the participants, comorbidities and pre-pandemic psychological conditions could help to understand the specific impact of the pandemic on mental health.
  • Multiple regression analysis is only shown for the psychological feelings, whereas the other pandemic variables only are analyzed using descriptive data. The association between phycological concerns and other variables could also provide relevant information.
  • The discussion section could be improved by reporting the relation between the main findings.
  • It would be interesting to analyze the influence of the government measures on the participant responses.
  • The limitations of the study should be preferably added to the discussion section.
  • The format and style of the manuscript need to be carefully and attentively revised. References are not in the journal format.

Round 2

Reviewer 2 Report

I appreciate the authors' effort to revise the manuscript  ijerph-1430916. However, they have only partially addressed some of the comments and suggestions and the work contains important methodological flaws.

Therefore, several aspects still need to be considered:

- Abstract:  the information related to the description of the sample, such as the age, along with the time in which the data were collected should be placed after the objective. The statistical significance of some results i.e. p values have not been included.

- Introduction: Neither the purpose of the study nor the study’s main contribution are presented at the final part of the introduction (including the literature review), just before the methods section, as suggested. The duration of COVID-19 counter-measures in the UAE and Abu Dhabi, as well as the specified timeframe concerning the development of the study.

Methods: This section now reports the questions included in the survey. However, the Socio-demographical characteristics of the participants still appear in this section instead of the Results section. The following statement has not been addressed: "The data analysis should specifically mention the type of statistics used as well as the software employed to perform the analysis". Moreover, the main and secondary objectives are included in this section and should be placed just before the method’s section, as mentioned above.

- Results and discussion: the following aspects still need further consideration.

  • As mentioned above, the sociodemographic table should appear in the Results section.
  • Although the data displayed in Table 1, Table 3, and Table 4 refer to descriptive data, the utilization of either parametric or not parametric statistics may inform on the characteristics and distribution of the sample.
  • Several recommendations have not been conveniently addressed and are postponed for future research. At least, the statistical analysis of results suggested for studying the association between phycological concerns and other variables should be considered.
